# Neutralizing antibodies reveal cryptic vulnerabilities and interdomain crosstalk in the porcine deltacoronavirus spike protein

Wenjuan Du[1,5], Oliver Debski-Antoniak [1,5], Dubravka Drabek [2,3], Rien van Haperen[2,3], Melissa van Dortmondt[1], Joline van der Lee [1], Ieva Drulyte [4], Frank J. M. van Kuppeveld [1], Frank Grosveld [2,3], Daniel L. Hurdiss [1] ✉ & Berend-Jan Bosch [1] ✉

Porcine deltacoronavirus (PDCoV) is an emerging enteric pathogen that has recently been detected in humans. Despite this zoonotic concern, the antigenic structure of PDCoV remains unknown. The virus relies on its spike (S) protein for cell entry, making it a prime target for neutralizing antibodies. Here, we generate and characterize a set of neutralizing antibodies targeting the S protein, shedding light on PDCoV S interdomain crosstalk and its vulnerable sites. Among the four identified antibodies, one targets the S1A domain, causing local and long-range conformational changes, resulting in partial exposure of the S1B domain. The other antibodies bind the S1B domain, disrupting binding to aminopeptidase N (APN), the entry receptor for PDCoV. Notably, the epitopes of these S1B-targeting antibodies are concealed in the prefusion S trimer conformation, highlighting the necessity for conformational changes for effective antibody binding. The binding footprint of one S1B binder entirely overlaps with APN-interacting residues and thus targets a highly conserved epitope. These findings provide structural insights into the humoral immune response against the PDCoV S protein, potentially guiding vaccine and therapeutic development for this zoonotic pathogen.

Coronaviruses (CoVs) constitute a diverse group of enveloped, single-stranded RNA viruses with the ability to infect mammals and birds. They are classified into four genera: *Alphacoronavirus*, *Betacoronavirus*, *Gammacoronavirus*, and *Deltacoronavirus*[1]. Within the *Betacoronavirus* genus, notable members include severe acute respiratory syndrome CoV (SARS-CoV), Middle East respiratory syndrome CoV (MERS-CoV), and SARS-CoV-2 which recently emerged in humans from animal reservoirs[2]. These events serve as poignant reminders of the capability of CoVs to cross species boundaries, thus posing a constant threat to human health.

The SARS-CoV outbreak began in November 2002 in the Guangdong province of China[3–5]. The virus likely originated in bats and was transmitted to humans[6–8], through intermediate hosts such as palm civets and racoon dogs[9,10]. It led to approximately 8000 cases worldwide, with a 10% mortality rate before containment in 2003. A decade later, in 2012, MERS-CoV emerged in Saudi Arabia[11]. While it has not established sustained human infections, it is recurrently re-emerging from its reservoir, the dromedary camel[12,13]. This virus has thus far been reported in over 2600 individuals, with around 35% of these cases succumbing to the infection. SARS-CoV-2 emerged in December 2019

[1]Virology Section, Infectious Diseases and Immunology Division, Department of Biomolecular Health Sciences, Faculty of Veterinary Medicine, Utrecht University, Utrecht, The Netherlands. [2]Department of Cell Biology, Erasmus Medical Center, Rotterdam, The Netherlands. [3]Harbour BioMed, Rotterdam, The Netherlands. [4]Thermo Fisher Scientific, Materials and Structural Analysis, Eindhoven, The Netherlands. [5]These authors contributed equally: Wenjuan Du, Oliver Debski-Antoniak. ✉e-mail: d.l.hurdiss@uu.nl; b.j.bosch@uu.nl

in Wuhan, China[14]. The virus is closely related to bat coronaviruses[15–20] and is thought to have jumped to humans through an intermediate animal host[21,22], although the exact host remains uncertain[23]. SARS-CoV-2, which caused the COVID-19 pandemic, has been devastating, causing more than 771 million reported infections and over 6.9 million deaths globally, up to November 2023 (covid19.who.int).

Porcine deltacoronavirus (PDCoV), classified within the genus *Deltacoronavirus*, was first discovered in pigs in Hong Kong in 2012[24], though its origin remains elusive. Since its initial detection, PDCoV outbreaks have surged among swine populations across various countries worldwide[25–30]. The virus infects intestinal epithelial cells and causes acute watery diarrhea, vomiting and dehydration in piglets, and can lead to death in nursing piglets[31–33]. Symptomatic PDCoV infection has also been observed in chickens, turkeys and bovine calves in experimental settings[34–36], indicating that the virus has a wide host range potential. Notably, in 2021, infection with PDCoV was documented in in plasma samples taken from three Haitian children presenting with acute febrile illness[37]. This finding highlights the potential for PDCoV to traverse from swine to human populations, thus emphasizing the need for vigilance and monitoring to curb potential transmission events.

The initial step of CoV infection is the engagement of the viral spike (S) protein with specific receptors on the host cell's surface[38]. The CoV S protein forms homotrimers and is a type I membrane protein[39]. It comprises two subunits: the N-terminal S1 subunit, responsible for receptor binding, and the C-terminal S2 subunit, which facilitates membrane fusion. The S1 subunit can be further divided into four core domains, namely S1[A–D], of which domains A and B are of importance in receptor binding. The S proteins of some CoVs (SARS-CoV, MERS-CoV and SARS-CoV-2) can adopt different conformations with the S1B domain either buried ("closed" or "down") or exposed ("open" or "up"), with the latter enabling the recognition of cellular receptors[40–43]. The aminopeptidase N (APN) has been identified as entry receptor for PDCoV and is bound by the S1B domain (also referred to as the C-terminal domain, CTD, Fig. 1A)[44,45]. Interestingly, PDCoV can utilize APN from different species, including humans, felines and chickens[44], highlighting its capacity for interspecies transmission. Furthermore, domain S1A (also known as the N-terminal domain, NTD, Fig. 1A) of the S protein has been reported to exhibit an affinity for sialic acid[46–48], implying that sialoglycans may serve as attachment factors for PDCoV entry into host cells.

Antibodies are important components of the humoral immune system against viral infection, and their potency as therapeutic agents is underscored by their pivotal role in countering COVID-19[49]. Virus-neutralizing antibodies target the S protein, and these S-specific antibodies have been harnessed to provide insight into the antigenic landscape of CoV S and evolutionary dynamics, as exemplified in studies on SARS-CoV-2[50–53], endemic human CoV OC43[54] and 229E[55]. However, despite the zoonotic potential of PDCoV, the precise epitopes and functional attributes of PDCoV-neutralizing antibodies remain unexplored.

In this study, we examine the antigenic structure of the PDCoV S protein. Through functional and structural characterization of a diverse panel of anti-PDCoV-S monoclonal antibodies (mAbs), we

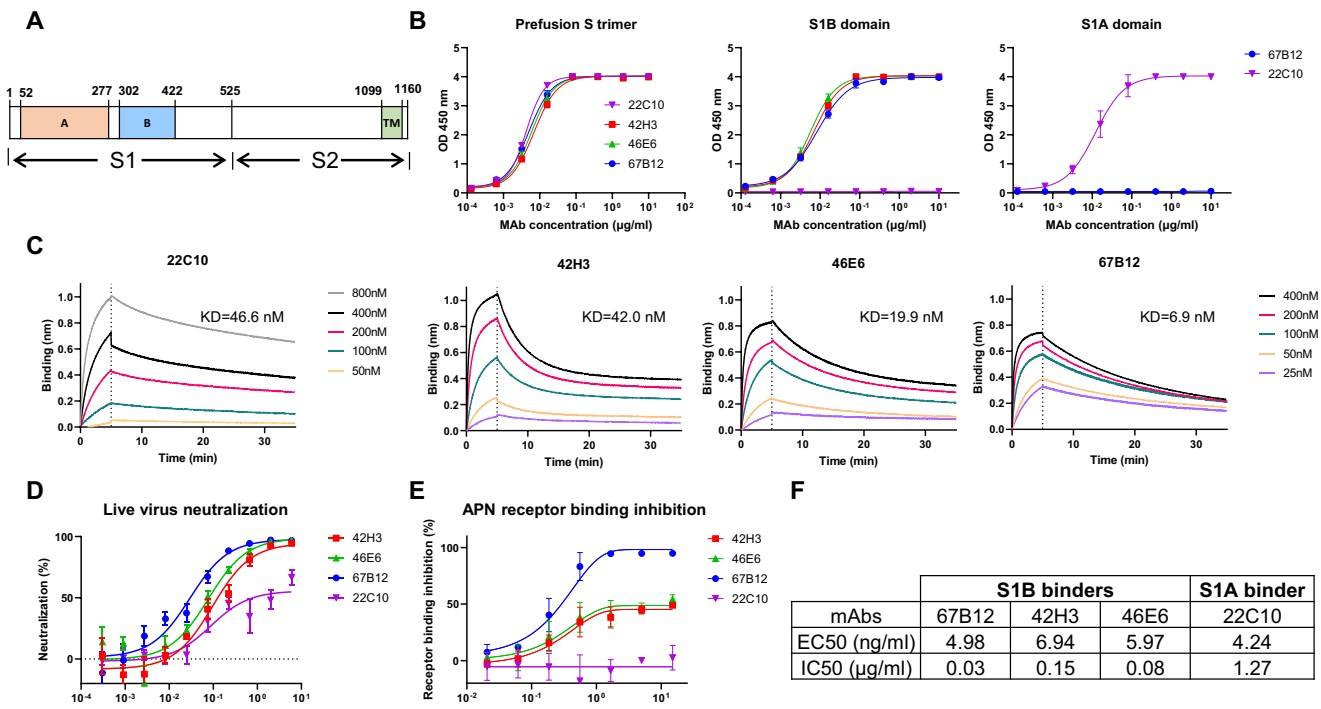

**Fig. 1 | Characterization of PDCoV neutralizing antibodies. A** Schematic representation of the PDCoV S protein, with domain A, domain B and the transmembrane domain (TM) labeled. **B** ELISA analysis showing mAb binding to immobilized prefusion PDCoV S trimer (left panel), S1B domain (middle panel) and S1A domain (right panel). **C** Binding kinetics of S-specific mAbs to PDCoV S1, measured through Bio-Layer Interferometry (BLI). Monoclonal antibodies were immobilized using anti-human Fc biosensors, and association and dissociation were observed with serially diluted PDCoV S1B, allowing calculation of equilibrium dissociation constants (KD). The experiment was conducted twice, with one representative experiment shown. **D** Neutralization of authentic PDCoV in Huh7 cells. PDCoV was preincubated with 3-fold serially diluted mAbs for 60 min before infecting Huh7

cells. Infection was quantified 15 h post infection by immunofluorescence microscopy. **E** ELISA-based receptor binding inhibition assay. PDCoV S1B, preincubated with serially diluted S mAbs, was added to a plate coated with soluble aminopeptidase N (APN). The interaction was quantified using HRP-conjugated antibody targeting the C-terminal Strep-tag fused to PDCoV S1B. Results represent the mean (±SD) from two independent experiments with at least two technical replicates. Source data are provided as a Source Data file. **F** EC50 (half-maximal effective concentrations) and IC50 (half-maximal inhibitory concentrations) values for each mAb calculated from the binding and neutralization curves displayed in (**B**) and (**D**), respectively.

provide insight into the neutralization of PDCoV and viral escape mechanisms. We structurally map vulnerable sites in domains A and B on the PDCoV S protein targeted by neutralizing antibodies and reveal that binders of S1B recognize concealed epitopes, thereby expanding our understanding of the dynamics of the PDCoV S protein. Taken together, these results provide structural insights for understanding humoral immune response against this zoonotic pathogen and hold promise as potent tools for outbreak preparedness against prospective PDCoV variants in humans.

## Results

### Identification of PDCoV S neutralizing mAbs

To elucidate the epitopes of neutralizing antibodies targeting the PDCoV S trimer, we developed hybridoma's derived from immunized H2L2 Harbour mice (https://www.harbourbiomed.com/) encoding chimeric immunoglobulins with human variable heavy and light chains and murine constant regions. The immunization strategy involved the use of plasmid DNA encoding the membrane-anchored PDCoV S protein and purified trimeric S ectodomain. Hybridoma supernatants were screened for reactivity against the PDCoV S trimer (Fig. S1A), and subsequently evaluated for neutralizing activity against authentic PDCoV, leading to the identification of four PDCoV neutralizing antibodies: 22C10, 42H3, 46E6 and 67B12 (Fig. S1B). The human heavy and light chain variable fragments of these antibodies were sequenced (Fig. S2) and human IgG1 antibodies were generated by cloning these regions into expression vectors encoding the human IgG1 heavy and kappa light chain constant regions. Human mAbs were recombinantly produced and purified, and further characterized for their binding specificity, in vitro neutralization, and receptor binding interference.

We performed ELISAs to determine the binding specificity of the selected human mAbs, which revealed that three mAbs: 42H3, 46E6 and 67B12 bound the APN-binding S1B domain, while mAb 22C10 bound to the S1A domain of the PDCoV S protein (Fig. 1B). In addition, using biolayer interferometry (BLI) we assessed cross-competition in binding of the three S1B-targeting mAbs to the S1 antigen. Antibodies 42H3 and 46E6 belonged to the same competition group while 67B12 targeted a distinct site on S1 (Fig. S3A). BLI analysis further demonstrated that all antibodies exhibited nanomolar binding affinity against monomeric S1 protein (Fig. 1C). Furthermore, we assessed the neutralization efficacy of the four antibodies against authentic PDCoV in a dose-dependent manner, determining their half maximal inhibitory concentration (IC50) values, which ranged from 0.03 to 1.27 μg/ml (Fig. 1D, F). To investigate the neutralization mechanism of these mAbs, we conducted an ELISA-based receptor binding inhibition assay. All three S1B binders inhibited binding of the S1B domain to APN, while the S1A binder 22C10 exhibited no interference with APN binding, as expected (Fig. 1E). Notably, 67B12 demonstrated complete inhibition efficiency, whereas 42H3 and 46E6 displayed only 50% inhibition at the highest mAb concentration (20 μg/ml). These results were consistent with a BLI-based receptor binding inhibition assay (Fig. S3B).

### Structural determination of 22C10 Fab bound to PDCoV spike

To gain insight into the interaction of mAb 22C10 with the PDCoV S1A domain, we conducted cryo-electron microscopy (cryo-EM) single-particle analysis of PDCoV S trimers incubated with the 22C10 Fab. Three-dimensional classification revealed a single class representing the prefusion S in a closed conformation. Consistent with our functional experiments, we observed clear density for 22C10 on the PDCoV S1A domain (Figs. 1A and 2A). Further processing yielded a reconstruction with a global resolution of 3.0 Å (Figs. S4 and S5). To further improve resolution at the epitope-paratope interface, which exhibited lower resolution due to flexibility of the complex, we conducted local classification and refinement focusing on the S1A domain and the Fab. This improved the local resolution to 3.1 Å, allowing for modeling of the entire Fab along with the interacting S region (Fig. 2B).

The interaction between 22C10 and the S antigen primarily involves the variable heavy (VH) chain of the antibody binding to the N-terminal region of the spike S1A. Notably, a hydrogen bond is formed between the variable light (VL) chain of 22C10 and residue H229 of S1A. The binding of 22C10 relies predominantly on hydrophobic interactions, further stabilized by VH hydrogen bonding with S1A domain residues S43, L45, Y46, T136, A137 and T138 alongside loop residues $^{229}$HLSA$^{232}$ (Fig. 2B). Importantly, the VH domain's interaction increased the stability to the spike's N-terminus, facilitating the extension of the previously reported spike structures by 8 amino acids through the formation of an α-helix (residues S43-N50)[56] (Fig. 2C).

Previously, it has been reported that removal of cell surface sialic acids reduces PDCoV infectivity[46,48], suggesting a potential interaction between sialoglycans and PDCoV S. However, our attempts to establish an S-sialoglycan, hemagglutination binding assay for assessing sialoglycan-binding interference by 22C10 were unsuccessful. We noted that the 22C10 bound-PDCoV S exhibits considerable conformational changes compared with the apo S, with a Root mean square deviation (RMSD) score of 2.8 Å, across the S protomer (Fig. 2C). Notably, the S1B domain undergoes an 11 Å shift towards a partially open conformation, increasing the solvent accessible surface area by 218 Å$^2$. However, based on BLI data, this 22C10-induced conformational shift is not sufficient to enable S interaction with APN (Fig. S3C). Indeed, when superimposing hAPN (PDB: 7VPQ) onto the 22C10-bound spike monomer, it is visually clear that the S1B is inaccessible (Fig. S3D).

To assess the conservation of the 22C10 epitope, we mapped the sequence variation in the S1A domain observed among field isolates onto the S1A-22C10 cryo-EM model (Fig. 2D). We observed considerable sequence variation at or near the 22C10 epitope (S44, L45, S231 and S234), suggesting that this epitope may be under immune pressure in nature. To further validate the neutralizing antibody epitope and to predict the effectiveness of 22C10 binding against variants, we expressed and analyzed binding to a set of S1 mutants. ELISA-based binding titration curves revealed that both L45A and Y46A mutations resulted in near complete loss of binding (Fig. 2E), confirming the significance of these residues in 22C10 binding. L45F, present in a recent human PDCoV isolate (Fig. S6), showed no change in 22C10 binding (Fig. 2E). L45H present in PDCoV field strains (Fig. S6), however, exhibited almost complete loss in 22C10 binding. Here, a polar residue has been substituted into a hydrophobic pocket, likely disrupting interactions in the pocket. The ΔN52 mutation, found in many Chinese PDCoV strains (Fig. S6) and close to the binding epitope of 22C10, had no effect on 22C10 binding.

To evaluate viral escape strategies from antibody-mediated neutralization, we performed serial passages of authentic PDCoV in the presence of escalating concentrations of 22C10 (Fig. S7). After five passages, we sequenced the S genes of the resultant escape mutant viruses and observed several mutations, including an S231F substitution and a deletion of three residues $^{232}$ANS$^{234}$. By ELISA, we observed that both mutations resulted in almost complete disruption of 22C10 binding (Fig. 2E). The loss of binding between the escape mutants can be attributed to structural changes. Specifically, the mutation S231F engenders a switch from a hydrophilic residue to a hydrophobic one, disrupting the hydrogen bond with the backbone of F101 that is responsible for the binding between these two residues. Additionally, the presence of phenylalanine, with its bulky side chain, likely causes steric hindrance, affecting multiple interactions between the $^{229}$HLSA$^{232}$ loop and 22C10. The deletion of $^{232}$ANS$^{234}$ similarly abolishes interactions that are focused on this loop through disruption of the epitope. Collectively, these data provide a structural understanding of antibody binding and escape for a neutralizing antibody targeting the proximal end of the PDCoV S1A domain.

### S1B neutralizers targeting cryptic epitopes in the closed S trimer

The three S1B neutralizers exhibited dose-dependent binding to the prefusion S trimer by ELISA (Fig. 1A), their binding to the prefusion S

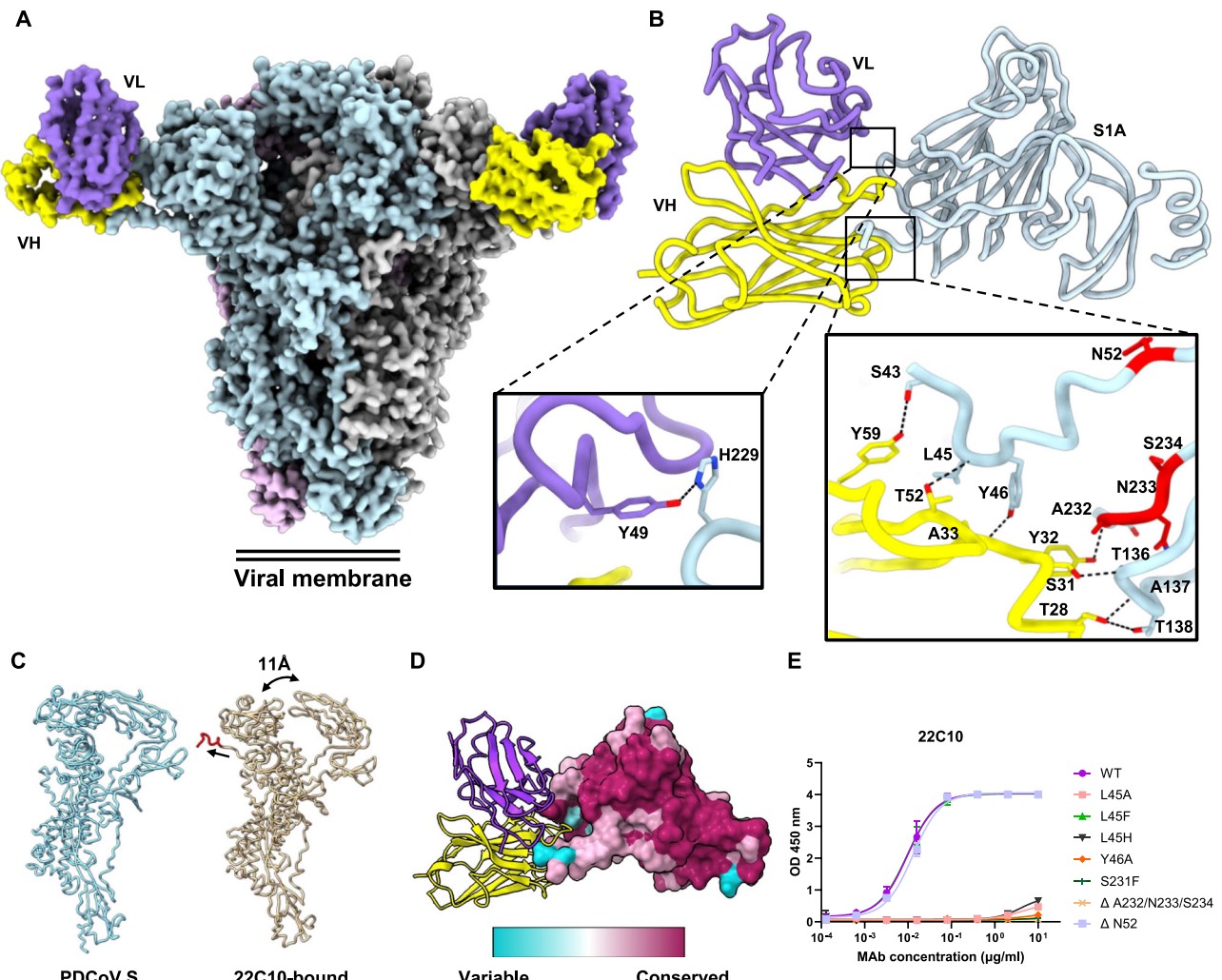

**Fig. 2 | Structural analysis of the 22C10 Fab fragment in complex with PDCoV S trimer. A** Surface representation of the trimeric PDCoV S bound to three 22C10 antibody Fab fragments. S protein protomers are in blue, gray, and pink, respectively. The Fab variable light chain is colored in purple and variable heavy chain labeled in yellow. **B** Atomic model of a single PDCoV S1A domain in complex with the 22C10 Fab fragment, including the interaction site of PDCoV S 22C10 Fab fragment. S1A is colored in blue and viral escape mutations highlighted in red. Fab variable light is colored in medium purple and variable heavy labeled in yellow.

**C** PDCoV S protomer comparison, left-PDCoV S protomer only and right-22C10 Fab fragment bound PDCoV S. The N-terminus, unresolved in the apo structure, is highlighted in red, and major architectural changes are marked with arrows.
**D** Mapping of PDCoV amino acid conservation onto the surface representation of PDCoV S1A domain, with the 22C10 Fab fragment bound. **E** ELISA binding reactivity of 22C10 to PDCoV S1 variants carrying the indicated mutations. Results represent the mean (±SD) from two independent experiments with two or three technical replicates. Source data are provided as a Source Data file.

trimer immobilized on BLI biosensor was not detected (Fig. 3A, middle panel). However, the S1A binder 22C10 retained its binding in this setup (Fig. 3A, middle panel). Interestingly, the S1B neutralizers demonstrated efficient binding to immobilized S1B monomer on the BLI biosensor (Fig. 3A, upper panel). These findings indicate that the epitopes targeted by S1B neutralizing antibodies remain concealed in the prefusion S trimer, and may be exposed due to partial disassembly of the trimeric S ectodomain during adsorption to the ELISA plates, as observed previously[54]. Further support for this observation comes from the inability of the three S1B neutralizers to bind cell surface expressed PDCoV S, unlike the 22C10 mAb (Fig. S8). These results further highlight the requirement for S protein conformational changes for effective binding which must be larger than observed in the slightly opened 22C10-bound S (Fig. S3C). Additional support for this hypothesis stems from the observation that these antibodies exhibited binding capacity to BLI biosensors loaded with heat-treated prefusion S trimers (Fig. 3A, lower panel). Heat treatment has previously been demonstrated to induce SARS-CoV-2 S conformational change[57], possibly to an open conformation in case of PDCoV S. Heat treatment may

also change the conformation of the S1A domain, resulting in the loss of 22C10 binding. Similarly to the S1B binders, APN also exhibited binding solely to the heat-treated prefusion S trimer (Fig. 3B). This finding underscores the importance of a conformational change of the S protein in enabling APN engagement, as previously suggested[56,58,59].

To address the challenge of the S1B mAbs inability to bind the prefusion S trimer, we aimed to determine the cryo-EM structure of the S1B domain in complex with the antibody Fab fragments. To achieve this, we combined the S1B domain with two non-competing Fabs, specifically the Fabs of 67B12 and 42H3 or 46E6. Of note, 42H3 and 46E6 share identical light chains and exhibit a minor variation of only five amino acids in the VH domain and as expected, show competitive binding to S1B (Fig. S3A). By forming these complexes, we increased the overall size of the complex, thus facilitating structural analysis through cryo-EM. Importantly, these complexes exhibited limited flexibility, enabling successful refinement and confident modeling of their structures. The cryo-EM reconstructions of the S1B domain complexed with 67B12/42H3 and 67B12/46E6 resulted in global resolutions of 3.0 Å and 2.9 Å (Figs. S9–S11).

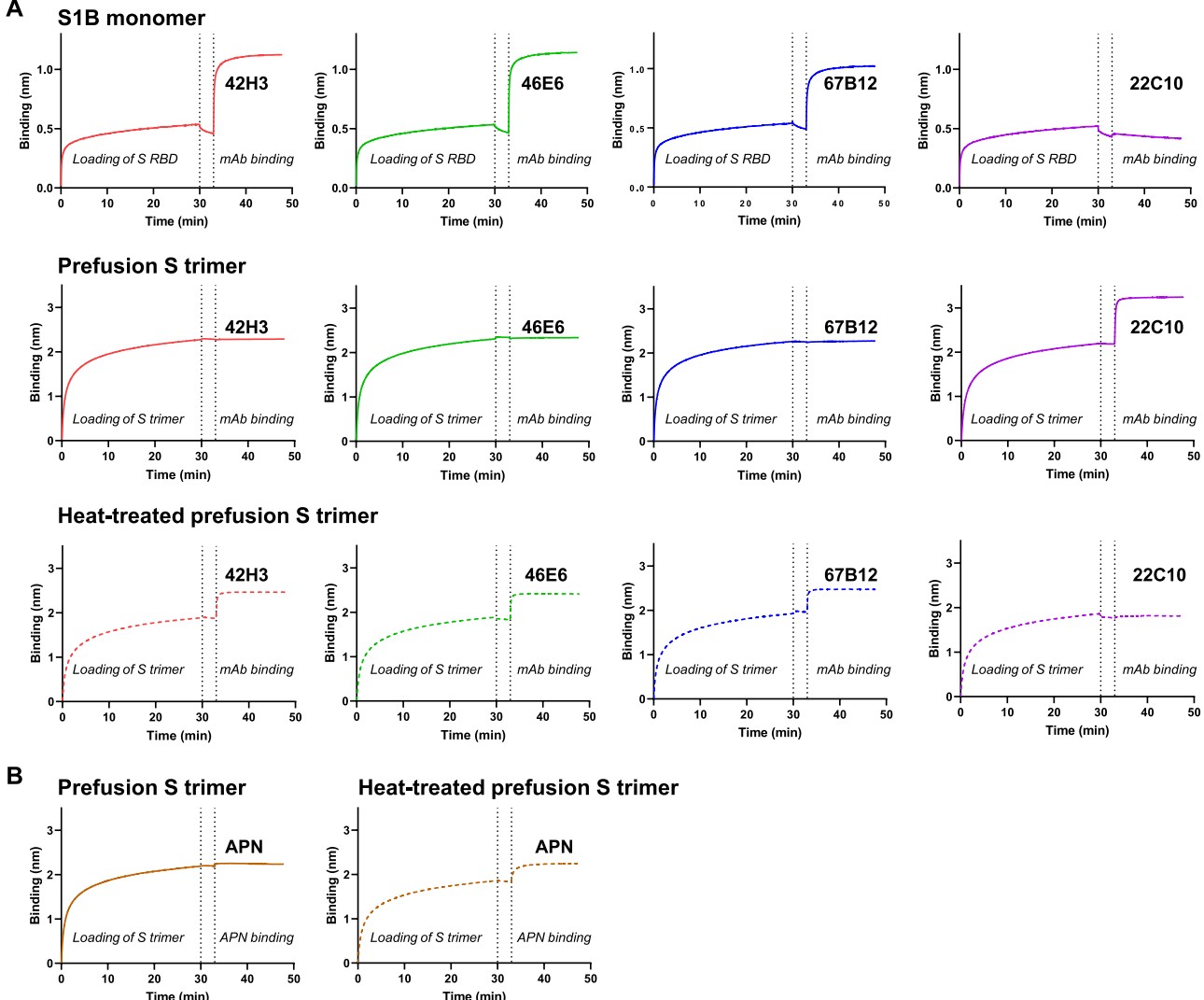

**Fig. 3 | Binding kinetics of antibodies and APN to S proteins using biolayer interferometry (BLI).** **A** Binding kinetics of S mAbs 42H3, 46E6, 67B12 and 22C10 to S RBD monomer (upper panel), prefusion PDCoV S trimer (middle panel), or to 60 °C-treated prefusion S trimer (lower panel). Antigens were immobilized onto the anti-strep mAb-coated Protein A BLI biosensors, followed by the association of each indicated mAb. **B** APN binding kinetics to non- (left panel) or 60 °C-treated (right panel) prefusion S trimer. Similar set-up was used as described under (**A**). Experiments were performed two times, one representative experiment is shown. Source data are provided as a Source Data file.

The cryo-EM structures of the S1B-Fab complexes offered insights into the specific binding regions of the antibodies. Both 67B12 and 42H3/46E6 were found to pack against each other, interacting with the ventral surface of the S1B domain (Fig. 4A). Consistent with our BLI data, these epitopes are inaccessible in the context of the prefusion trimeric spike (Fig. 4C, D).

The 67B12 mAb directly interacts with the receptor binding loops of the S1B domain, rationalizing its antiviral potency as it competes with the host-cell receptor APN (Figs. 4B and S3B). In contrast, 42H3/46E6 interact near the hinge region of S1B, which extends to form the S1C domain. These antibodies do not appear to interfere directly with the S1B-APN interaction. No significant shift in the backbone of PDCoV S1B or in the side chains, which interact with APN were observed when comparing S1B, APN-bound S1B and 42H3/46E6-bound S1B (Fig. S12). The antiviral potency of these antibodies may be explained by alternative mechanisms, such as leveraging/shedding of S1 from the S2 domain, preventing efficient receptor binding and ultimately entry into the host cell, as observed with neutralizing SARS-CoV-2 antibodies[60] (Fig. S13).

## Mapping specific interactions of S1B cryptic neutralizers and assessment of viral escape

For both complexes, the obtained resolution of the cryo-EM maps allowed us to interpret the interactions between S1B and 67B12/42H3/46E6 at the level of side chain orientations (Fig. 5A). Both 42H3 and 46E6, which differ by only five amino acids in the VH domain, exhibit similar interactions with S1B. Both Fabs form hydrogen bonds with specific S1B residues, including T360, S362, E387, and E410, alongside a salt bridge to E410, further stabilizing the Fab-S1B interaction. Notably, 42H3 exhibits a unique hydrogen bond with A408 of the S1B domain.

As mentioned above, 67B12 exhibits direct interactions with the receptor binding loops of S1B. Notably, several residues, including E320, R357, V395, W396, N397, R401, and R403 of the S1B domain, are involved in hydrogen bonds with 67B12. Additionally, there is an observed salt bridge formation between L399 of S1B and D103 of 67B12. The VH domain of 67B12 is primarily responsible for engaging with the target, while the VL domain interacts with S1B residue E320 via N92 located in CDR3.

To investigate the potential escape mechanisms of the 42H3 and 46E6 antibodies, we conducted serial passage experiments with

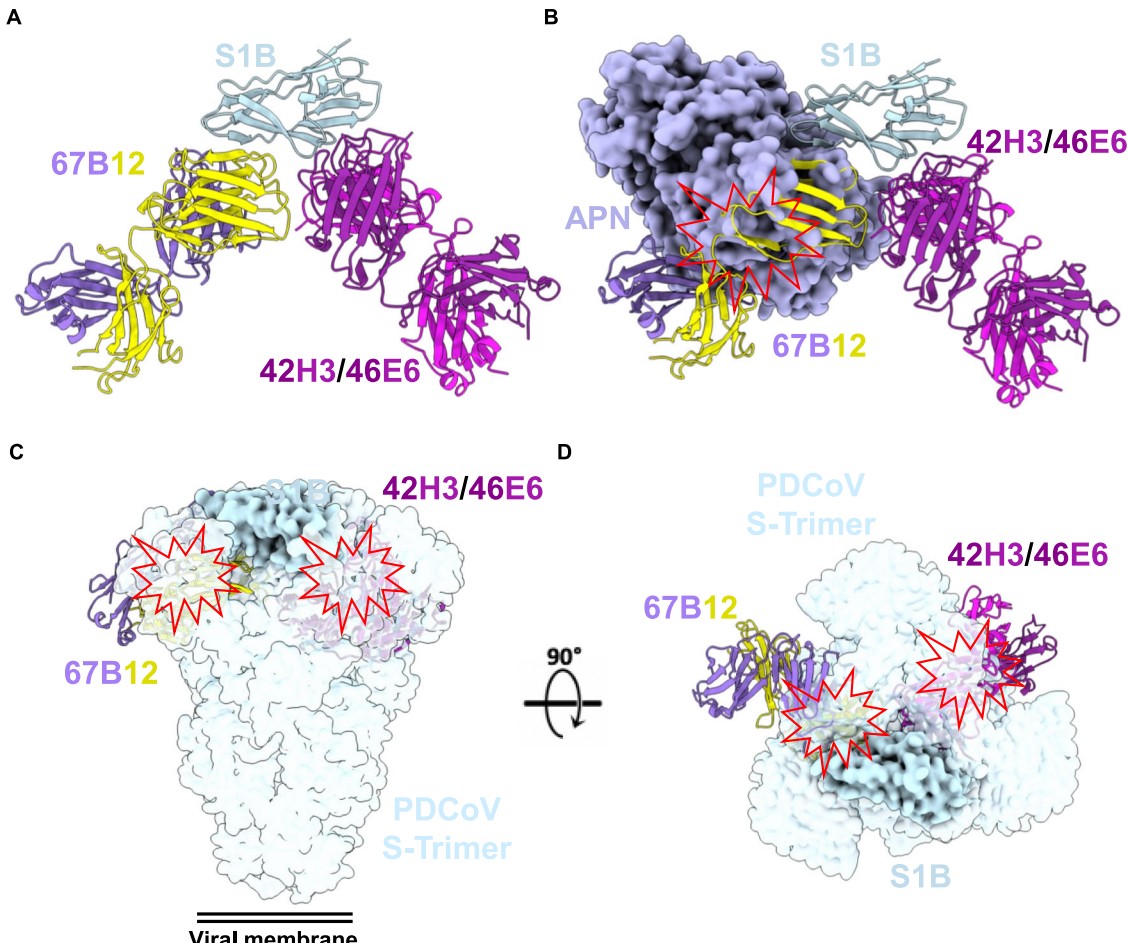

**Fig. 4 | Structures of neutralizing antibodies bound to S1B reveal cryptic epitopes in PDCoV S trimer. A** Atomic model representation of the PDCoV S1B (blue) bound to 67B12 antibody ab fragment (medium purple and yellow) and the 42H3/46E6 antibody Fab fragments (purple and pink). **B** The interaction of hAPN (surface representation-orange) with the S1B of PDCoV (PDB:7VPQ) overlayed onto the atomic structure of the S1B-Fab complexes. **C, D** S1B-Fab complexes overlayed onto the surface representative of the PDCoV S trimer displayed from two perpendicular viewpoints showing that the 67B12 and 42H3/46E6 epitopes are inaccessible in the closed trimer.

PDCoV under escalating concentrations of each antibody, following a strategy similar to that utilized for 22C10 (Fig. S7). To assess the impact of escape mutations on antibody binding, we conducted ELISA assays. Markedly, these mutations are directly involved in, or close to, the observed interactions. The S362R mutation, selected by both 42H3 and 46E6 antibodies, induced a significant reduction in binding for both antibodies, with a more pronounced effect on 46E6 (Fig. 5B, C). Similarly, the combined mutation D359A / E410Q, selected by 46E6, led to a substantial decrease in binding for both antibodies, while each individual mutation showed a moderate negative effect on binding. Moreover, the E410G mutation, selected by 42H3, noticeably impaired the binding of 42H3, with only a minor effect on 46E6 binding.

From a structural perspective (Fig. S14), the S362R mutation affects the binding of both 42H3 and 46E6 to M103. The bulky nature of the arginine may induce steric hindrance, resulting in a reduction in binding strength. This reduction is more pronounced in 46E6, as M103 represents the sole interaction with S362R in this complex. In contrast, 42H3 benefits from an additional residue, V104, which further stabilizes the interaction. Individually D359A and E410Q mutations have limited effects on binding. D359A does not directly interact with 42H3 or 46E6 but lies near critical interactions between S1B T360 and VH R105, as well as VL Y32 for both antibodies. While the E410Q mutation can still interact similarly to wild type, the switch prevents salt-bridge formation, weakening this interaction. Finally, differences in binding between 42H3 and 46E6 for the E410G mutation can be explained by

the number of interactions E410 forms with each Fab. 46E6 exhibits a single interaction, a salt bridge formed between E410 via R99. Conversely, 42H3 presents four interactions with S1B E410 via its VH domain (I101, Y32, R98 and R99), including the same salt-bridge formation with R99, making this a key interaction for 42H3. These interactions are disrupted due to the mutation from a bulky negatively charged side chain to the smallest neutral side chain.

The binding footprint of 67B12 on the surface of S1B overlaps with a high proportion of residues that contact APN during receptor binding site (RBS) engagement (44.4%) (Fig. 5F), accounting for its high neutralization potency. During this interaction, 702 Å² of the S1B solvent-accessible surface area is buried (~11% of the total), while the interaction with APN buries 853 Å² of the solvent-accessible surface area (~13% of the total), emphasizing the similarities of the 67B12 binding footprint to the PDCoV host-cell receptor. The significance of the identified interacting residues was evaluated by introducing alanine substitutions in S1 (Fig. S15B). Among the key residues, only the W396A mutation, positioned at the core of the binding epitope, displayed a clear reduction in binding to 67B12 when assessed using the ELISA format, although binding was not completely abolished (Fig. 5D), suggesting 67B12 interactions are highly stable and can endure single mutations in the binding epitope. Importantly, previous studies have demonstrated that this mutation abolishes both pAPN and hAPN binding[58], indicating the functional conservation of this residue. Indeed, analyzing the sequence variability within the S1B domain, we observe a high degree of

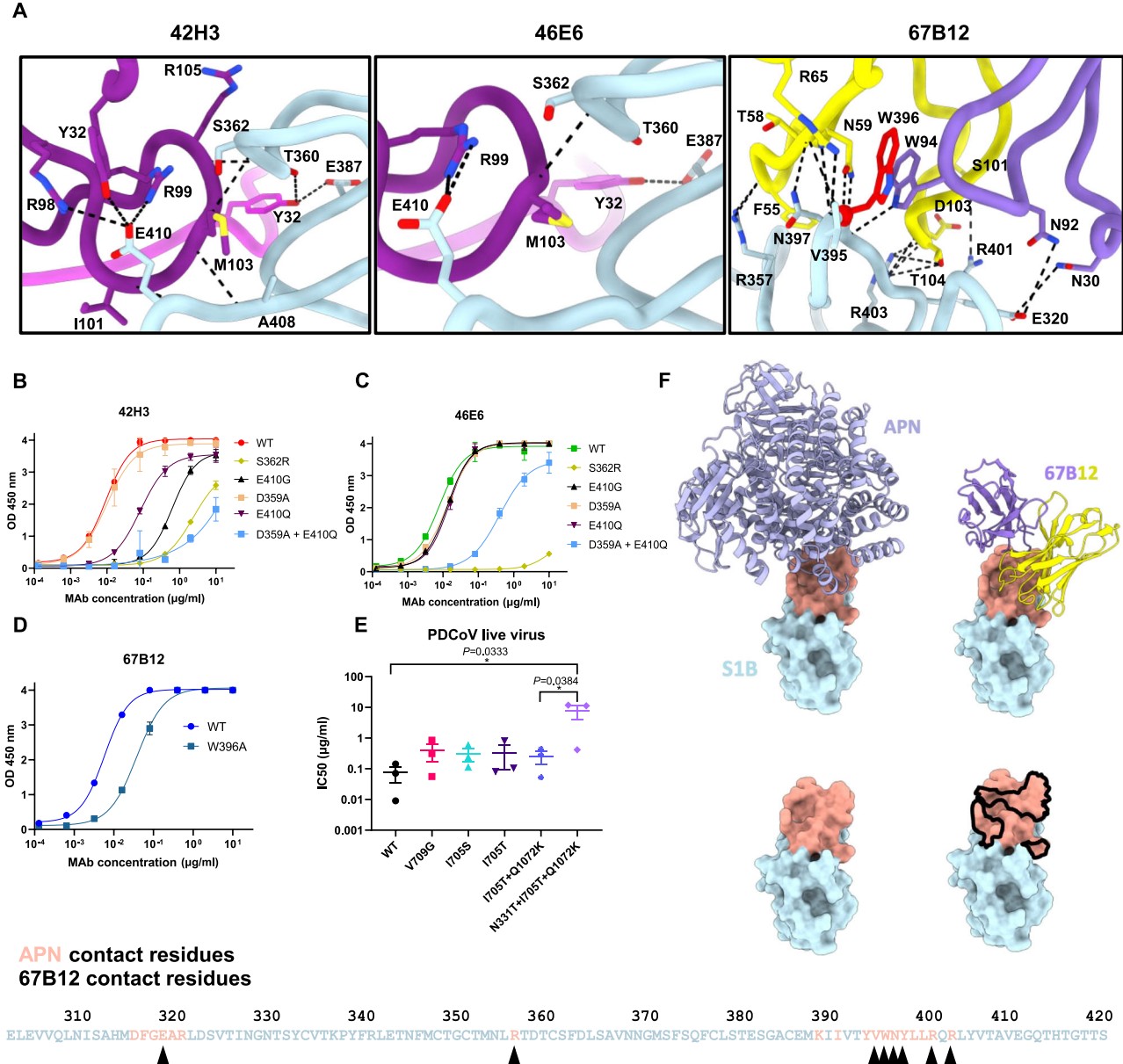

**Fig. 5 | Analysis of interactions between PDCoV S1B and S1B-targeting neutralizing antibodies guided by structural insights and escape mutations.**
**A** Interaction sites of S1B with Fab fragments 42H3, 46E6 and 67B12, respectively.
**B** ELISA-binding curves for 42H3 and (**C**) 46E6 to PDCoV S1 antigens containing (combinations of) single-site mutations. The experiments were performed twice with double technical replicates, the average ± SD is shown. **D** ELISA-binding curves for 67B12 to PDCoV S1 and the PDCoV S1 W396A mutant. Data points represent the average ± SD, for $n = 1$ replicate from two independent experiments. OD 450 nm: optical density measured at 450 nm. **E** IC50 values of 67B12-mediated

neutralization of PDCoV wildtype (WT) and variants carrying the indicated S mutations. Presented are the average ± SD of three biological replicates with three technical replicates. $p$ values were obtained using one-way ANOVA with Tukey's multiple-comparison test. Source data are provided as a Source Data file.
**F** Comparison of binding footprint (in peach) of human APN on PDCoV S1B (PDB: 7VPQ) with that of the antibody Fab fragment 67B12 (black outline). Underneath the PDCoV S1B domain sequence is shown with residues highlighted that contact APN (in peach) or 67B12 (arrow heads).

conservation of 67B12 binding epitope (Fig. S15A). With the exception of a single residue N397, a range of porcine PDCoV isolates exhibit K397 in the S1B domain (Fig. S6). However, this 397 K substitution had no effect on 67B12 binding (Fig. S15B).

To further substantiate this observation, we assessed the escape efficiency of PDCoV against 67B12. The virus underwent five passages with increasing concentrations of 67B12 (Fig. S7). Mutant viruses were isolated by limited dilution, propagated, and evaluated for their resistance to 67B12. Mutant viruses carrying either single mutations (V709G, I705S, I705T), double mutations (I705T + Q1072K) or triple mutations (N331T + I705T + Q1072K) were successfully obtained (Fig. S7). Most of the observed S mutations (V709G, I705S, I705T, I705T +

Q1072K) slightly reduced neutralization by 67B12, although not significantly (~3-fold increase in IC50; Fig. 5E). Only the combination of three mutations (N331T + I705T + Q1072K) significantly reduced 67B12-mediated neutralization, although S1B mutation N331T had no effect on 67B12 binding (Fig. S15B). These escape mutations are positioned away from the epitope and did not completely abolish the neutralization by 67B12. We hypothesize that the combination of these mutations is likely to impede conformational changes required for the PDCoV prefusion S to adopt an "open" S1B conformation necessary for binding to the host cell receptor APN. Notably, the observed I705S, I705T and V709G mutations reside in the fusion-peptide proximal region (FPPR, residues L689-Q712) of the S protein, which is situated

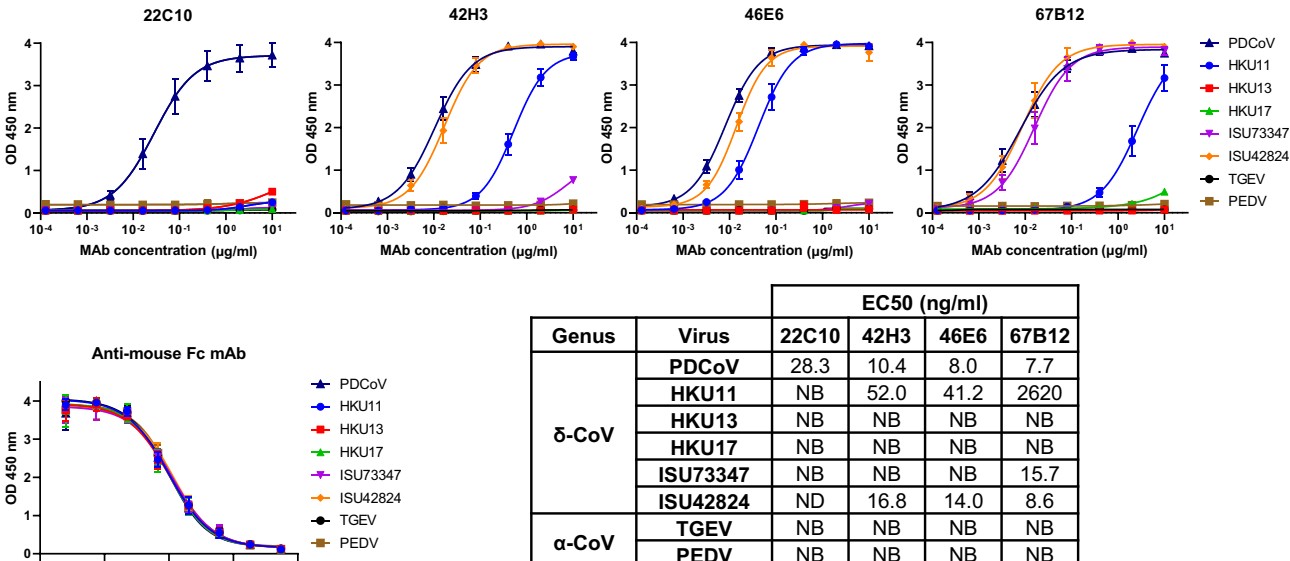

**Fig. 6 | Binding breadth of PDCoV S1-directed mAbs.** ELISA-binding curves of the S1A mAb 22C10 and the S1B mAbs 42H3, 46E6, 67B12 to S1 antigens (S1B in case of ISU42824) of indicated coronaviruses from the deltaCoV and alphaCoV genus, coated at equimolar concentrations. An anti-mouse Fc mAb was used to corroborate equimolar plate coating of the mouse Fc-tagged S antigens. Symbols represent mean values ± SD from two independent experiments, with two technical replicates. The ELISA-based half-maximal effective concentrations (EC50) titers are shown in the table. NB no binding, ND not determined. Source data are provided as a Source Data file.

between subdomains S1C and S1D. Similarly, the FPPR of SARS-CoV-2 (SARS-CoV-2 S residues 828–853) functions as a key regulator of the open and closed sampling of the RBD, through its interaction with subdomains S1C and S1D (Fig. S16)[61,62]. Therefore, mutations in this structural motif may modulate S conformational dynamics and consequently narrow the vulnerability window in which 67B12 can interact with the exposed epitope.

## Cross-reactivity of PDCoV S mAbs to other CoVs

To assess their cross-binding capabilities, we screened the ELISA-based reactivity screening of the four neutralizing monoclonal antibodies against S1 antigens from various deltaCoVs, including Bulbul coronavirus HKU11-934, Munia coronavirus HKU13-3514, three sparrow coronaviruses HKU17, ISU73347 and ISU42824[63], as well as the alpha-CoVs transmissible gastroenteritis virus (TGEV) and porcine epidemic diarrhea virus (PEDV). While the S1A binder 22C10 showed no binding to any tested S antigens (Fig. 6), we observed cross-reactivity of the three S1B mAbs with spike antigens of some of the deltaCoVs. Notably, 67B12 displayed comparable binding kinetics with sparrow coronaviruses ISU73347 and ISU42824, and additional, albeit lower, binding to HKU11-934. The degree of conservation in spike protein contact residues among the deltaCoVs supports the breadth of binding observed with the S1B binders (Fig. S17), indicating that these S1B mAbs target a conserved epitope shared among deltaCoVs.

## Discussion

Despite its recognition as a zoonotic pathogen, the antigenic landscape of PDCoV remains inadequately understood. To address this knowledge gap, we generated and characterized a range of PDCoV S-targeting neutralizing mAbs. Through an integrated approach involving functional and structural analyses, we successfully pinpointed vulnerable regions within the prefusion S trimer. Our efforts unveiled three distinct epitopes: one situated on the S1A domain, and two epitopes on the S1B domain that are concealed within the closed, prefusion S trimer. Moreover, by challenging authentic PDCoV with these mAbs, we identified mutations which may aid in understanding viral escape pathways.

Our S1A-directed mAb, 22C10, induces conformational changes in the S trimer upon binding, thereby bringing the S1B domain towards a partially open conformation, though this movement was still insufficient for APN binding. While merbeco- and sarbecovirus prefusion S proteins exhibit dynamic sampling of open and closed conformations[40–43], most other known coronaviruses predominantly display closed conformations[56,64–68]. Uniquely, alphacoronavirus HCoV-229E S protein has been reported to adopt a partially open conformation[69], analogous to conformational changes observed during 22C10 binding to PDCoV S. Recent findings demonstrate that the serotype A HKU1 S undergoes large-scale conformational changes upon binding a primary sialoglycan receptor via domain S1A, resulting in the exposure of the S1B domain[70]. This conformational shift enables host-cell receptor binding and entry, revealing an intriguing immune evasion mechanism in CoVs. Our results with 22C10 suggest a similar crosstalk between the S1A and S1B domains in PDCoV. It is conceivable that 22C10 binding may disrupt the S protein's interaction with an unidentified host-cell co-receptor, potentially by obstructing or overlapping the co-receptor binding site. In support of this hypothesis, PDCoV has been shown to interact with sialic acids through the S1A domain[48].

The discovery of cryptic epitopes of S1B-directed neutralizing antibodies in this study adds to a growing list of CoVs utilizing this immune evasion strategy. Previous work from our lab has identified similar vulnerable epitopes in the OC43 S[54], suggesting the presence of conformational flexibility and the possibility of a putative secondary receptor. Additionally, epitopes occluded in the closed S trimer state have been identified for SARS-CoV-2, only becoming accessible by neutralizing antibodies when the receptor-binding domain is in the open conformation[51]. These findings may indicate a common mechanism utilized by CoVs to evade immune detection and neutralization.

Although the 42H3/46E6 epitope does not overlap with the APN binding site, the binding of 42H3/46E6 to S1B interferes with receptor binding. This interference is unlikely due to glycan clashes, as glycosylation sites are distant from the antibody epitope. Instead, it's plausible that these antibodies induce subtle allosteric shifts in the RBD loops, reducing APN binding efficiency. These antibodies bind

closely to the S1C domain, which acts as a hinge to enable S1B to adopt the open conformation. This interaction possibly represents the primary mechanism by which these antibodies neutralize PDCoV, potentially inducing S1 shedding or prefusion S disruption, similar to core-binding SARS-CoV-2 neutralizing antibodies[71–74].

Among the identified candidates, 67B12 shows promise for future therapeutic use as it demonstrates cross-reactivity against related avian-origin deltacoronaviruses and its binding footprint on S1B overlaps completely with residues that contact with APN. In addition, we did not observe escape mutations that reduce 67B12 binding to S1B, indicating that such mutations may compromise viral fitness loss due to their impact on APN engagement. Instead, we observed mutations within regions of S1B and S2 that are positioned away from the 67B12 epitope, suggesting an allosteric immune escape mechanism.

In summary, the identification of neutralizing epitopes on PDCoV S, as presented in this study, provides a structural framework to understand the humoral immune response against the spike protein of this zoonotic pathogen. This may inform vaccine and therapeutic development, monitoring emerging variants, and advancing our broader understanding of (zoonotic) diseases.

## Methods

### Viruses and cells
Huh7 cells were cultured in Dulbecco's modified Eagle's medium (DMEM) supplemented with 10% FBS, sodium pyruvate (1 mM; Gibco), 1x nonessential amino acids (Lonza), penicillin (100 IU/ml) and streptomycin (100 IU/ml) at 37 °C in a humidified $CO_2$ incubator. The cell line was tested and confirmed negative for mycoplasma contamination. The PDCoV virus was purchased from the US Department of Agriculture and propagated and titrated on Huh7 cells in DMEM supplemented with 10% FBS.

### Expression and purification of PDCoV S protein
The human codon-optimized gene encoding the PDCoV S ectodomain (residues 1–1098, GenBank: AHL45007.1) with a C-terminal GCN4 trimerization motif, followed by a strep-tag for purification was synthesized by GenScript. PCR fragments for PDCoV-S1A (amino acid 20-297) and PDCoV-S1B (amino acid 298-425) were cloned in frame between the CD5 signal peptide and a triple strep-tag for purification. These proteins were expressed transiently in HEK-293T [American Type Culture Collection (ATCC), CRL-11268] cells using pCAGGS expression plasmids and purified from culture supernatants using streptactin beads (IBA) according to the manufacturer's protocol. Single-site residue substitutions in S1 variants were generated by site-directed mutagenesis using Q5 High-fidelity DNA polymerase (NEB).

The gene fragments encoding S1 of PDCoV (amino acid 1-525), HKU11-934 (amino acid 1-565, GenBank: YP_002308479), HKU13-3514 (amino acid 1-555, GenBank: YP_002308506), HKU17 (amino acid 1-569, GenBank: YP_005352846), ISU73347 (amino acid 1-524, GenBank: AWV67134), TGEV (amino acid 1-781, GenBank: AGM75126), PEDV (amino acid 1-728, GenBank: AFP81695) and S1B of ISU42824 (amino acid 296-422, GenBank: AWV67125) were synthesized by GenScript. All gene fragments were cloned into the pCAGGS expression vector in frame with the mouse Fc and produced from HEK-293T cells as described above. These proteins were purified using Protein A Sepharose (IBA) according to the manufacturer's instructions.

### Immunization, hybridoma culturing and production of recombinant mAbs
Immunization of Harbour H2L2 transgenic mice with appropriate PDCoV S expression plasmids and proteins, hybridoma fusion, screening and sequencing of human variable regions was done as previously described for SARS-CoV-2 S[75]. Recombinant human antibodies were produced in HEK-293T cells. Briefly, VH and VL chain sequences of each antibody were synthesized by Genscript and cloned into expression

plasmids with human IgG1 heavy chain and κ chain constant regions, and expressed in HEK-293T cells using transient transfection. At 18 h after transfection, the transfection mixture was replaced by 293 SFM II expression medium (Invitrogen), supplemented with sodium bicarbonate (3.7 g/liter), glucose (2.0 g/liter), Primatone RL-UF (3.0 g/liter), penicillin (100 IU/ml), streptomycin (100 IU/ml), GlutaMAX, and 1.5% dimethyl sulfoxide. Tissue culture supernatants were harvested 5 days after transfection, from which human antibodies were purified using Protein A Sepharose (IBA) according to the manufacturer's instructions.

### Antibody binding to PDCoV S antigens analyzed by ELISA
Purified Strep-tagged S antigens (1 µg/ml) were coated onto 96-well NUNC Maxisorp plates (Thermo Fisher Scientific) at 4 °C overnight, followed by three washing steps with phosphate-buffered saline (PBS) containing 0.05% Tween 20. Alternatively, purified mouse Fc-tagged S1 or S1B antigens with equal molarity (1.2 pM) were used for coating. Plates were then blocked with 3% bovine serum albumin (BSA; Fitzgerald) in PBS with 0.1% Tween 20 at RT for 2 h. Antibodies in hybridoma supernatants, diluted in PBS containing 3% BSA and 0.1% Tween 20, were incubated with the S antigen-coated ELISA plates at RT for 1 h, and the binding was determined using a 1:3000 diluted horseradish peroxidase (HRP)-conjugated mouse anti-rat IgG1, IgG2b, and IgG2 mix (Absea; Catalog# KT96/KT98/KT99) for 1 h at RT. Alternatively, human PDCoV S mAbs were incubated with the plates at fivefold serial dilutions, starting at 10 µg/ml diluted in PBS containing 3% BSA and 0.1% Tween 20, at RT for 1 h. Antibody binding to the S proteins was detected using a 1:2000 diluted HRP-conjugated goat anti-human IgG (ITK Southern Biotech; Catalog# 2040-05) for 1 h at RT. Coating efficiency of mouse Fc-tagged proteins was detected using 2-fold serially diluted HRP-conjugated goat anti-mouse IgG (Invitrogen; Catalog# 31432) with the starting concentration of 3.2 µg/ml. HRP activity was measured at 450 nm using tetramethylbenzidine substrate (BioFX) and an ELISA plate reader (EL-808, BioTek).

### Antibody affinity determination using biolayer interferometry
Human PDCoV S mAbs (20 nM) were loaded onto Protein A biosensors (ForteBio) for 10 min. Subsequently, the biosensors were dipped into wells containing recombinant PDCoV S1 monomer (starting concentration: 800 nM for 22C10 and 400 nM for other three antibodies) with twofold dilutions for 5 min. Baseline was achieved between each step through incubation in PBS for 3 min. A long dissociation step (30 min) was carried out to observe the decrease in the binding response. The affinity constant KD was calculated using 1:1 Langmuir binding model on Fortebio Data Analysis 7.0 software. The experiment was carried out at 30 °C.

### Biolayer interferometry-based binding competition assay
Binding competition was performed using biolayer interferometry (Octet Red348; ForteBio). In brief, PDCoV-S1 or S1B domain (50 µg/ml) was immobilized onto the anti-strep mAb-coated protein A biosensor. After a brief washing step, the biosensor tips were immersed into a well containing primary mAb (mAb1; 50 µg/ml) for 10 min or 30 min and subsequently into a well containing the competing mAb (mAb 2; 50 µg/ml) for 10 min or recombinant soluble APN (100 µg/ml) for 30 min. A 3-min washing step in PBS was included in between steps. The experiment was carried out at 30 °C.

### ELISA-based receptor binding inhibition assay
The receptor binding inhibition assay was performed as described previously[75]. Briefly, the 96-well NUNC Maxisorp (Thermo Fisher Scientific) plate was coated with 1 µg chicken APN per well at 4 °C overnight[44], followed by 2 h blocking step at room temperature (RT) in blocking buffer (3% BSA in PBS containing 0.1% Tween 20). Three-fold serially diluted antibodies with a starting concentration of 15 µg/ml in blocking buffer were incubated with 2 nM PDCoV-S1B at RT for 1 h.

Afterwards the mixture was added to the plate and incubated for 2 h at 4 °C, after which the plate was washed 3 times with PBS containing 0.05% Tween. The binding of the S1B to APN was detected by HRP-conjugated StrepMab-classic (IBA; Catalog# 2-1509-001; 1:2000) via its binding to affinity purification Strep-tag fused to S1B. Detection of the HRP activity was quantified using tetramethylbenzidine substrate (BioFX) and an ELISA reader at 450 nm. The percentage of inhibition was calculated as the ratio of the reduction in binding in the presence of mAbs normalized to the binding in the absence of mAbs.

### Binding kinetics of mAbs and APN to PDCoV S proteins using biolayer interferometry (BLI)

Recombinant monomeric S1B, trimeric S or pre-heated trimeric S (heated at 60 °C for 10 min, followed by incubating at 4 °C for 10 min) was immobilized onto the anti-strep mAb-coated protein A biosensor as described above (loading of S). After a 3-min washing step, the biosensor tips were dipped into a well containing either PDCoV S mAb (mAb binding, 50 μg/ml) or APN (100 μg/ml) for 30 min.

### Immunofluorescence assay

Huh7 cells were transfected with a pCAGGS expression plasmid encoding the full length PDCoV S protein. Twenty-four hours after transfection, cells were washed twice with PBS and fixed with 3.7% formaldehyde (Merck) in PBS for 30 min at room temperature. Fixed cells were then blocked using 3% BSA in PBS for 1 h, followed by incubation with the indicated, 5-fold serially-diluted PDCoV S monoclonal antibody, starting at 10 μg/ml. After three washes, staining of cells was completed by incubating cells with goat anti-human Alexa Fluor 488 antibody (Invitrogen; Catalog# A-11013; 1:400). Nuclei were visualized using DAPI nuclear counterstaining (D-9542, Sigma). Pictures of immunofluorescent cells were captured using an EVOS M5000 cell imaging system (Thermo Fisher Scientific) at ×4 magnification.

### Live virus neutralization assay

Three-fold serially diluted mAbs with the starting concentration of 6 μg/ml were incubated with authentic PDCoV (600 TCID50) at RT for 1 h before infection of Huh7 cells. After 16-h infection, the cells were fixed with 3.7% formalin, and permeabilized with 0.1% Triton-100. PDCoV-infected cells were stained with mouse anti-PDCoV nucleocapsid antibodies (Clinisciences; Catalog# PDCOV11-M; 1:2000) followed by donkey anti-mouse Alexa Fluor 488 antibody (Invitrogen; Catalog# A-21202; 1:400). Pictures of immunofluorescent cells were taken using an EVOS M5000 cell imaging system (Thermo Fisher Scientific) at ×4 magnification and analyzed using ImageJ. Neutralization (%) was calculated as the ratio of the reduction in infected cells in the presence of mAbs normalized to mock-infected cells. The IC50 values were determined using four-parameter logistic regression (GraphPad Prism v10).

### Generation and characterization of neutralization escape mutants

A three-fold serially diluted range of each mAb (starting concentration of 6 μg/ml) was pre-incubated with an equal volume of PDCoV (600TCID50) at RT for 1 h, and then transferred to 24-well plate with fully confluent monolayer of Huh7 cells, and incubated at 37 °C. At 72-h post infection, cell culture supernatants containing progeny virus were harvested for wells with the highest antibody concentration showing clear cytopathic effect (CPE). Subsequent passages in the presence of increased concentration of mAb were generated by using virus containing supernatant from previous passage from the highest concentration of the antibody that allowed the observation of CPE. This process was continued up to passage 5. Viral RNA at passage five from 100 μl virus containing supernatant was isolated using the NucleoSpin RNA virus kit (Macherey-Nagel), followed by RT-PCR (Invitrogen). The S genes of individual antibody-treated virus were sequenced by Sanger sequencing.

For isolation of clonal escape mutant virus, virus aliquots of passage 5 were two-fold serially diluted before laying over fully confluent monolayers of Huh7 cells. Cell culture supernatants from wells that still displayed clear CPE at the highest dilution of the virus were harvested 72 h post infection. The S genes of each clonal virus were sequenced as described above to confirm the virus carrying specific mutations.

### Cryo-electron microscopy sample preparation and data collection

For the spike-22C10 complex, 4.3 μl of PDCoV S-ectodomain, at a concentration of 10.5 μM (based on the molecular weight of the spike protomer) was combined with 0.2 μl of 320 μM 22C10 Fab and incubated for ~10 min at room temperature.

For S1B-67B12/42H3 Fab complex and S1B-67B12/46E6 Fab complex, the components of each complex were pre-incubated at room temperature for 10 min prior to co-purification on a Superose 6 increase 10/300 column using UV absorbance at 215 nm on AKTA Pure system (GE Healthcare) running in tris-buffered saline (TBS) buffer. The fractions containing S1B-Fab complexes were concentrated using 10 kDa cut-off Amicon ultrafiltration units.

For all samples, immediately before blotting and plunge freezing, 0.5 μl of 0.1% (w/v) fluorinated octyl maltoside (FOM) was added to the sample, resulting in a final FOM concentration of 0.01% (w/v).

Subsequently, all complexes were treated identically. The material (3 μl) was applied to glow-discharged (20 mAmp, 30 s, Quorum Glo-Qube) Quantifoil R1.2/1.3 grids (Quantifoil Micro Tools GmbH), blotted for 5 s using blot force 0 and plunge frozen into liquid ethane using Vitrobot Mark IV (Thermo Fisher Scientific). The data were collected on a Thermo Fisher Scientific Krios™ G4 Cryo Transmission Electron Microscope (Cryo-TEM) equipped with Selectris X Imaging Filter (Thermo Fisher Scientific) and Falcon 4i Direct Electron Detector (Thermo Fisher Scientific) operated in Electron-Event representation (EER) mode.

In total, 3014, 4736 and 5808 movies were collected for 22C10, S1B-67B12/42H3 & S1B-67B12/46E6 Fab complexes, respectively, at a nominal magnification of ×165,000, corresponding to a calibrated pixel size of 0.73 Å/pix over a defocus range of −0.75 to −1.5 μm. A full list of data collection parameters can be found in Table S1.

### Single particle image processing

Data processing was performed using the CryoSPARC Software package[76]. After patch-motion and CTF correction, particles were picked using a blob picker, extracted at 3x (Spike-22C10) or 4x (S1B-67B12/42H3 & S1B-67B12/46E6 Fab complexes) binning and subjected to 2D classification. Following 2D classification, particles belonging to class averages that displayed high-resolution detail were selected for ab-initio reconstruction into five classes. Particles belonging to the whole representative complex class were re-extracted at 1.4x (Spike-22C10) or 1.3x (C S1B-67B12/42H3 & S1B-67B12/46E6 Fab complexes) x binning. Here, the spike-22C10 complex was treated differently to both S1B-67B12/42H3 & S1B-67B12/46E6 Fab complexes).

The spike-22C10 complex was subjected to non-uniform refinement with optimization of per-particle defocus and per-group CTF parameters[77]. At this point, the global resolution of the complex was 3 Å, however, the epitope interface between spike and Fab was resolved to a lower resolution. To improve local resolution, particles in the final C3 global reconstruction were symmetry expanded, a custom mask encompassing one S1A domain of the spike with bound 22C10 Fab was used to carry out a 3D variability job, in which particles assigned to the class with the best detail in the paratope-epitope were selected. Selected particles were then used in cryoSPARC local refinement (BETA). This markedly improved local resolution, with the epitope resolved to a resolution of 3.1 Å, enabling sufficient confidence for modeling this epitope.

S1B complexes were treated similarly from here on out. Following re-extraction, hetero refinements into five classes were run for both

complexes to remove "junk" particles from the final reconstructions. The class representing the highest resolution data was then taken forward, and a non-uniform refinement with optimization of per-particle defocus and per-group CTF parameters[77] was carried out, achieving resolutions of 3.0 Å and 2.8 Å, respectively, with sufficient information to build atomic models confidently. For a more detailed processing methodology, see Figs. S4, S9 and S10.

## Model building and refinement

UCSF Chimera[78] (version 1.15.0) and Coot[79] (version 0.9.6) were used for model building. The structure of the PDCoV spike glycoprotein previously resolved (PDB ID 6BFU)[56] and AlphaFold2 generated 22C10 Fab[80,81] was used as a starting point for modeling of the spike-22C10 complex. AlphaFold2 multimer was used to generate S1B-67B12/42H3 Fab complex, which was later mutated at key residues of Fab 42H3, to create S1B-67B12/46E6 Fab complex. Models were individually rigid body fitted into the density map using the UCSF Chimera "Fit in map" tool and then combined. The resulting model was then edited in Coot using the 'real-space refinement, carbohydrate module[82] and 'sphere refinement' tool. To improve fitting, Namdinator[83] was utilized, using molecular dynamics flexible fitting of all models. Following this, iterative rounds of manual fitting in Coot and real space refinement in Phenix[84] were carried out to improve rotamer, bond angle and Ramachandran outliers. During refinement with Phenix, secondary structure and non-crystallographic symmetry restraints were imposed. The final model was validated in Phenix with MolProbity[85], EMRinger[86], and fitted glycans validated using Privateer[87,88].

## Structure analysis and visualization

Interacting residues of PDCoV spike-Fab epitopes were identified using PDBePISA[89] and LigPlot+[90] Figures were generated using UCSF ChimeraX[91]. Structural biology applications used in this project were compiled and configured by SBGrid[92].

## Statistics and reproducibility

Statistical significance was performed using GraphPad Prism software (v10). One-way analysis of variance (ANOVA) with Tukey's multiple-comparison test was applied for multiple comparisons with one independent variable. A $p$ value of less than 0.05 was considered significant. All functional experiments were repeated at least twice on different days, and similar results were obtained. No statistical method was used to predetermine sample size. No data were excluded from the analyses.

## Reporting summary

Further information on research design is available in the Nature Portfolio Reporting Summary linked to this article.

## Data availability

The PDB file of PDCoV spike protein (PDB ID: 6BFU) and SARS-CoV-2 spike protein (PDB ID: 6XR8) were downloaded from NCBI database (https://www.ncbi.nlm.nih.gov/). Spike protein sequences used in this study were downloaded from NCBI database (https://www.ncbi.nlm. nih.gov/) (see Supplementary Fig. 6 or "Methods" section for the accession numbers). The cryo-EM maps and atomic structures have been deposited in the Protein Data Bank (PDB) Electron Microscopy Data Bank (EMDB) under accession codes: 8R9W and EMD-19014 for PDCoV S with 22C10 (global), 8R9X and EMD-19015 for PDCoV S with 22C10 (local refinement), 8R9Y and EMD-19016 for PDCoV 67B12/42H3 Fab complex, and 8R9Z and EMD-19017 for PDCoV 67B12/46E6 Fab complex. Sequences of the monoclonal antibodies described here are available from GenBank under the following accession numbers: PP886668 (22C10 VL), PP886669 (22C10 VH), PP886670 (42H3/46E6 VL), PP886671 (42H3 VH), PP886672 (46E6 VH), PP886673 (67B12 VL) and PP886674 (67B12 VH). Data underlying Figs. 1b–e, 2e, 3a, b, 5b–e, 6 and Supplementary Figs. 1a, 3a–c, 15b have been deposited in a publicly accessible repository (https://figshare.com/articles/dataset/PDCoV_S_mAb/25533358). Source data are provided with this paper.

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

## Acknowledgements

We thank Tony Smits for technical support and Wentao Li for providing DCoV S1 constructs. This study was done within the framework of the research programme of the Netherlands Centre for One Health (www.ncoh.nl). This work was partially funded by the Corona Accelerated R&D in Europe (CARE) project. The CARE project has received funding from the Innovative Medicines Initiative 2 Joint Undertaking (JU) under grant agreement No. 101005077 (to B.J.B. and F.J.M.K.). The JU receives support from the European Union's Horizon 2020 research and innovation program, the European Federation of Pharmaceutical Industries Associations (EFPIA), the Bill & Melinda Gates Foundation, the Global Health Drug Discovery Institute, and the University of Dundee. The content of this publication reflects only the authors' views, and the JU is not responsible for any use that may be made of the information it contains.

## Author contributions

Experiment design: W.D., O.D.-A., D.L.H., and B.J.B.; gene cloning, protein expression, and purification: W.D., M.D., and J.D.L.; isolation and characterization of monoclonal antibodies: W.D., M.D., J.D.L., D.D., R.H., and F.G.; binding and neutralization assays: M.D. and J.D.L.; cryo-EM data collection, processing, and model building: O.D.-A., I.D., and D.L.H.; bioinformatic analysis of virus genetic diversity: W.D. and B.J.B.; antibody escape mutant selection and sequencing: W.D., M.D., and J.D.L.; data analysis: W.D., O.D.-A., D.L.H., and B.J.B.; supervision: F.J.M.K., F.G., D.L.H., and B.J.B.; study conception and coordination: B.J.B.; manuscript writing: O.D.-A., W.D., D.L.H., and B.J.B., with input from all other authors.

## Competing interests

D.D., R.H., and F.G. are employees of Harbour Biomed and hold company shares. I.D. is an employee of Thermo Fisher Scientific. The remaining authors declare no competing interests.
