## [Peer Review File · Nature Communications]

Neutralizing antibodies reveal cryptic vulnerabilities and interdomain crosstalk in the porcine deltacoronavirus spike proteinREVIEWER COMMENTS

Reviewer #1 (expert in molecular characterisation of viral and bacterial proteins):

This manuscript from Du et al reports the isolation and characterization of four antibodies that bind the spike protein from porcine deltacoronavirus (PDCoV). This virus mainly infects pigs although recently was detected in several humans, causing acute febrile illness in three Haitian children. The four antibodies were obtained by immunizing H2L2 Harbour mice, resulting in antibodies with human variable domains. All four antibodies were shown to neutralize PDCoV, and mapping studies determined that three bind the S1B (or RBD) domain and one binds the S1A (or NTD) domain. High-resolution cryo-EM structures were obtained for each antibody in complex with the PDCoV spike (22C10) or isolated S1B domain (42H3, 46E6 and 67B12), allowing for a precise definition of the epitope-paratope interactions. Mutagenesis studies validated the structures, as did the selection of antibody-escape variants. Of the four antibodies, 67B12 is the most potent, and directly competes with binding of the APN receptor. 67B12 was also difficult to select resistance variants for, and only a triple-substitution variant significantly reduced neutralization. This antibody may therefore have some utility as a medical countermeasure, should it be needed.

The manuscript and studies within are thorough, and the experiments are performed to a high standard. The cryo-EM data processing is sound, and the model statistics are excellent. A weakness of the manuscript is that the mechanism of neutralization for three of the antibodies is unknown. The authors suggest that 42H3 and 46E6 may neutralize by inducing shedding of the S1 domain. The manuscript would be strengthened if the authors could perform this experiment and test their hypothesis.

Minor comments:

1. Line 238 states "The four S1B neutralizers" but this should read "The three S1B neutralizers"
2. The authors should add a footnote to Table S1 explaining to a general reader what the values in parentheses indicate for the R.m.s. deviations.

Reviewer #2 (expert in porcine coronavirus):

PDCoV has a broad host range and could infect human as well, thus posing potential threat to public health. The antigenic landscape of PDCoV is currently lacking. In this manuscript, Du et al generated four neutralizing mAbs with human variable regions from transgenic mice. The cryoEM structures of PDCoV S ectodomain or RBD bound to Fab of these mAbs were solved to atomic resolution, which allowed for reliable binding analysis. Via targeting S1A domain, mAb 22C10 introduces conformational change on S protein and leads to partial exposure of RBD. The Fabs of 67B12, 42H3 and 46E6 all bind to cryptic epitopes in RBD that are inaccessible in the prefusion trimeric S protein. Specifically, the epitope of 67B12 largely overlaps with residues in S protein responsible for binding to APN, which indicates that 67B12 neutralizes PDCoV infection by preventing receptor binding. In addition to structural work, the authors also carried out detailed biochemical and reverse genetic studies to validate epitope of these antibodies. This work is of considerable significance and provides potential therapeutic candidates against PDCoV. However, there are some issues needed to be addressed.

- 1) The specificity of the generated mAbs has not been shown. Do they cross-react with spike proteins from other coronaviruses, such as PEDV, TGEV and other deltacoronaviruses?
- 2) In line 197-198, the authors claim that 22C10 induced conformation change on S protein is not sufficient to fully expose RBD for APN binding. However, there is not data in Figure S3 to support this claim.
- 3) Is there binding competition between APN and antibodies to heat-treated prefusion S trimer?
- 4) The four antibodies all bind to S protein prefusion trimer in the ELISA assay (Fig. 1B), while in BLI assay, only the binding between 22C10 and S protein has been detected (Fig. 3A). The authors should address this discrepancy.

Some minor comments:

- 1) In Fig. 2C and Fig. 5A (right panel), too much structure detail was squeezed into one panel. To improve readability, the authors could split these into different zoom-in views.

- 2) In lines 324-338, the authors tried to explain the mechanism of viral escape under the selection pressure of antibodies from the structural perspective. They should provide supplementary figures to show hydrogen bonding / salt bridges mentioned in the manuscript.
- 3) To further validate the resolution of the cryoEM map, the authors may provide the density of the binding interface with fit model, for example, in Fig. S5.
- 4) The nomenclature of domains in S protein should be consistent. There are S1B and RBD, which refer to the same domain, throughout the manuscript.
- 5) Line 42, Betacoronavirus genus, not Betacoronaviruses genus
- 6) Line 238, there are only three S1B neutralizers, not four.
- 7) Line 239, prefusion S trimer, not prefusion S timer
- 8) Fig. 3B, "APN binding" is mislabeled to "mAb binding" in the BLI sensogram.

Reviewer #3 (expert in Structural biology):

Du et al. presented the isolation and analysis of multiple neutralizing antibodies targeting the spike glycoprotein (S) of PDCoV, a deltacoronavirus with zoonotic implications. The methods used in this study included measuring binding, competing with receptors, neutralizing live virus, elucidating antibody epitopes by complex structure determination, and testing viral immune escape mutations in the binding interface. These findings enhance our knowledge of PDCoV S protein dynamics and offer structural insights into the humoral immune response to this zoonotic pathogen.

One limitation of this study is the lack of exploration into the antibody's working mechanism. For the 42H3/46E6 antibodies, authors are suggested to compare the structures of the S1B (RBD) in apo-, APN-binding and antibody-binding states to explore potential allosteric shifts in the RBD loops. For the possible shedding mechanism, experimental data is also needed.

Figure 1D indicates that 22C10 only neutralized the live virus infection by 50%, unlike the other three RBD-targeting antibodies. Is this a plateau, or can the concentration of 22C10 be increased further to achieve greater neutralization? Interestingly, the authors noted that 22C10 binding to the S1A (NTD) induces conformational changes in the S by causing the S1B to move towards a partially open conformation. This raises the question of whether 22C10 could have a synergistic effect with the other three RBD-targeting antibodies.

Could the authors please explain why the ELISA method could detect the binding of three RBD-targeting antibodies to the prefusion S, while the BLI method could not?

It is also noteworthy that in Figure 3A, the heated S lost its binding with 22C10, but gained binding with three RBD-targeting antibodies. Could the authors explain the loss of 22C10 binding?

In line 238, the number of S1B neutralizers is three, not four.

In line 110, Binding kinetics of S-specific mAbs to PDCoV S1B. S1B or S1?

RESPONSE TO REVIEWERS' COMMENTS

We would like to thank all Reviewers for their accurate summaries of our results and constructive criticism. We have performed additional analyses to support the findings of this study. Below, we provide detailed responses to the reviewers' questions and suggestions, as indicated in italics.

Reviewer #1 (expert in molecular characterization of viral and bacterial proteins):

This manuscript from Du et al reports the isolation and characterization of four antibodies that bind the spike protein from porcine deltacoronavirus (PDCoV). This virus mainly infects pigs although recently was detected in several humans, causing acute febrile illness in three Haitian children. The four antibodies were obtained by immunizing H2L2 Harbour mice, resulting in antibodies with human variable domains. All four antibodies were shown to neutralize PDCoV, and mapping studies determined that three bind the S1B (or RBD) domain and one binds the S1A (or NTD) domain. High-resolution cryo-EM structures were obtained for each antibody in complex with the PDCoV spike (22C10) or isolated S1B domain (42H3, 46E6 and 67B12), allowing for a precise definition of the epitope-paratope interactions. Mutagenesis studies validated the structures, as did the selection of antibody-escape variants. Of the four antibodies, 67B12 is the most potent, and directly competes with binding of the APN receptor. 67B12 was also difficult to select resistance variants for, and only a triple-substitution variant significantly reduced neutralization. This antibody may therefore have some utility as a medical countermeasure, should it be needed.

We thank the reviewer for the detailed and constructive assessment of our manuscript. We have used the valuable feedback to improve the overall presentation of the manuscript.

The manuscript and studies within are thorough, and the experiments are performed to a high standard. The cryo-EM data processing is sound, and the model statistics are excellent. A weakness of the manuscript is that the mechanism of neutralization for three of the antibodies is unknown. The authors suggest that 42H3 and 46E6 may neutralize by inducing shedding of the S1 domain. The manuscript would be strengthened if the authors could perform this experiment and test their hypothesis.

While we acknowledge that more insight into the mechanism of neutralization of the 42H3 and 46E6 antibodies would enhance the manuscript, conducting experiments to test hypotheses – including the S1 shedding hypothesis – presents challenges due to the inaccessibility of the epitopes targeted by these S1B mAbs in the prefusion trimeric spike ectodomain, as demonstrated by BLI (Fig.3) and cryo-EM analysis (Fig.4). In our efforts to address this, we attempted to test their binding to full-length PDCoV spike on the plasma membrane, by staining spike-transfected Huh7 cells. However, in contrast to the S1A-targeting 22C10, all three S1B binders failed to bind to the cell surface-expressed spike, confirming the cryptic nature of their epitopes also in the full-length, membrane-embedded spike protein (Fig.S8). Consequently, we

infer that the cryptic epitopes targeted by these neutralizing antibodies are only transiently exposed – presumably during virus entry, as we also outlined in the manuscript (lines 246-256). Given the challenges of testing the 42H3 and 46E6 neutralization mechanism experimentally, we opted to investigate this through structural modelling. Through the superposition of the open PEDV spike (PDB: 7Y6V) onto our 22C10-bound PDCoV spike, we generated plausible models for the 1-up and 2-up trimer. Subsequently, we aligned our 67B12 and 42H3 complex to the open RBD to see if clashes with the adjacent S protomers would be observed. As shown in the new supplementary figure 13, 67B12 could engage the open PDCoV spike without any steric hindrance. 42H3, on the other hand, would clash with the adjacent S protomers in both the 1-up and 2-up states. For SARS-CoV-2, such clashes are indicative of mAbs which induce spike destabilization, such as observed for CR3022 (PMID: 32585135). As such, we believe that these models support our hypothesis that these core targeting mAbs could destabilize the PDCoV spike trimer.

Minor comments:

1. Line 238 states “The four S1B neutralizers” but this should read “The three S1B neutralizers”

This has been corrected.

2. The authors should add a footnote to Table S1 explaining to a general reader what the values in parentheses indicate for the R.m.s. deviations.

Thank you for identifying this issue. The values in the parentheses were added accidentally and are unnecessary. These have now been removed.

Reviewer #2 (expert in porcine coronavirus):

PDCoV has a broad host range and could infect human as well, thus posing potential threat to public health. The antigenic landscape of PDCoV is currently lacking. In this manuscript, Du et al generated four neutralizing mAbs with human variable regions from transgenic mice. The cryoEM structures of PDCoV S ectodomain or RBD bound to Fab of these mAbs were solved to atomic resolution, which allowed for reliable binding analysis. Via targeting S1A domain, mAb 22C10 introduces conformational change on S protein and leads to partial exposure of RBD. The Fabs of 67B12, 42H3 and 46E6 all bind to cryptic epitopes in RBD that are inaccessible in the prefusion trimeric S protein. Specifically, the epitope of 67B12 largely overlaps with residues in S protein responsible for binding to APN, which indicates that 67B12 neutralizes PDCoV infection by preventing receptor binding. In addition to structural work, the authors also carried out detailed biochemical and reverse genetic studies to validate epitope of these antibodies. This work is of considerable significance and provides potential therapeutic candidates against PDCoV. However, there are some issues needed to be addressed.

We thank the reviewer for the constructive suggestions. In particular, we thank the reviewer for prompting us to test the neutralization breadth of our mAbs. These new data further highlight the potential of our molecules for pandemic preparedness against deltacoronaviruses.

1) The specificity of the generated mAbs has not been shown. Do they cross-react with spike proteins from other coronaviruses, such as PEDV, TGEV and other deltacoronaviruses?

Thank you for raising the question regarding the specificity of the generated monoclonal antibodies and their potential cross-reactivity with spike proteins from other coronaviruses. To address this, we conducted ELISA binding assays to evaluate the binding of all four mAbs to spike proteins from the alphacoronaviruses PEDV and TGEV, and various avian-origin deltacoronaviruses including Bulbul coronavirus HKU11-934, Munia coronavirus HKU13-3514, three sparrow coronaviruses HKU17, ISU73347 and ISU42824. While the S1A-targeting 22C10 exhibited no binding to any tested S antigens, we observed cross-reactivity of the S1B mAbs to spike proteins of some of the avian deltacoronaviruses, consistent with the conserved nature of their epitopes. We have incorporated the cross-binding data into the revised manuscript (lines 381-402, Fig.6).

2) In line 197-198, the authors claim that 22C10 induced conformation change on S protein is not sufficient to fully expose RBD for APN binding. However, there is not data in Figure S3 to support this claim.

We tested whether the partial opening of the spike trimer induced by 22C10 exposure is sufficient to enable APN binding using BLI. Our new data confirm that the conformational shift induced by 22C10 binding is not adequate to facilitate S interaction with APN. We have included these data in the manuscript (lines 196-197 and Fig.S3C), alongside an additional figure, which provides a structural representation of the clash that occurs when APN is superimposed onto the 22C10-bound PDCoV spike structure (lines 197-199 and Fig.S3D).

3) Is there binding competition between APN and antibodies to heat-treated prefusion S trimer?

Our experiments, based on the results from the ELISA (Fig.1E) and BLI assay (Fig.S3B), suggest that receptor engagement by PDCoV S1B is completely inhibited by 67B12 and to a lesser extent by 42H3 and 46E6. We did not perform the equivalent experiments with the heat treat spike as this is a rather harsh method to expose the S1B domain and likely does not faithfully mirror natural conformational changes which occur in the spike during cell entry. As such, we do not believe these experiments would yield much more meaningful insights than the S1B competition experiments.

4) The four antibodies all bind to S protein prefusion trimer in the ELISA assay (Fig. 1B), while in BLI assay, only the binding between 22C10 and S protein has been detected (Fig. 3A). The authors should address this discrepancy.

We included an explanation of this phenomenon in the Results section of the revised manuscript, specifically in lines 242-245. We hypothesize that the observed discrepancy may arise from the partial disassembly of the metastable prefusion S trimers during adsorption to the ELISA plates, which could potentially expose the epitopes targeted by the antibodies. We have noted a similar phenomenon for antibodies that target domain-occluded epitopes in the OC43 spike trimer

(PMID: 35614127). Furthermore, our new data showing the inability of the three S1B neutralizers to bind cell surface expressed PDCoV S (unlike the 22C10 mAb) provides additional support for the cryptic epitope nature of the antibodies in question (lines 245-247 and Fig.S8).

Some minor comments:

1) In Fig. 2C and Fig. 5A (right panel), too much structure detail was squeezed into one panel. To improve readability, the authors could split these into different zoom-in views.

Thank you for this suggestion. We agree these panels were too crowded, and we have simplified them to show only key interactions. Additionally, we have split the views in Fig.2C into two separate zoom-in views. We also took the opportunity to make stylistic changes to Figures 2 and 4, which we believe are now much clearer.

2) In lines 324-338, the authors tried to explain the mechanism of viral escape under the selection pressure of antibodies from the structural perspective. They should provide supplementary figures to show hydrogen bonding / salt bridges mentioned in the manuscript.

We included a supplementary figure (Figure. S14) to address viral escape mechanisms from 42H3 and 46E6 from a structural perspective.

3) To further validate the resolution of the cryoEM map, the authors may provide the density of the binding interface with fit model, for example, in Fig. S5.

We included representative densities for each cryo-EM map/model (Fig.S5 and Fig.S11).

4) The nomenclature of domains in S protein should be consistent. There are S1B and RBD, which refer to the same domain, throughout the manuscript.

We replaced all instances of "RBD" with "S1B" throughout the manuscript.

5) Line 42, Betacoronavirus genus, not Betacoronaviruses genus

This has been corrected.

6) Line 238, there are only three S1B neutralizers, not four.

This has been corrected.

7) Line 239, prefusion S trimer, not prefusion S timer

This has been corrected.

8) Fig. 3B, "APN binding" is mislabeled to "mAb binding" in the BLI sensogram.

This has been corrected.

Reviewer #3 (expert in Structural biology):

Du et al. presented the isolation and analysis of multiple neutralizing antibodies targeting the spike glycoprotein (S) of PDCoV, a deltacoronavirus with zoonotic implications. The methods used in this study included measuring binding, competing with receptors, neutralizing live virus, elucidating antibody epitopes by complex structure determination, and testing viral immune escape mutations in the binding interface. These findings enhance our knowledge of PDCoV S protein dynamics and offer structural insights into the humoral immune response to this zoonotic pathogen.

We thank the reviewer for the constructive suggestions. We have used the valuable feedback to improve the overall presentation of the manuscript.

One limitation of this study is the lack of exploration into the antibody's working mechanism. For the 42H3/46E6 antibodies, authors are suggested to compare the structures of the S1B (RBD) in apo-, APN-binding and antibody-binding states to explore potential allosteric shifts in the RBD loops. For the possible shedding mechanism, experimental data is also needed.

We have explored potential allosteric shifts in the RBD loops in the S1B (RBD) in apo-, APN-binding and antibody-binding states. We have taken your recommendation and overlaid the various S1B/RBDs of the respective structures: Apo-spike, 22C10-bound, 42H3 –bound and APN-bound. No significant shift in the backbone of PDCoV S1B or in the side chains, which interact with APN was observed when comparing S1B, APN-bound S1B and 42H3/46E6-bound S1B (lines 286-288, Fig.S12) indicating that allosteric shifts in the RBD loops are unlikely to be the cause of inhibition.

In response to your recommendation regarding the shedding mechanism, we kindly refer to the response provided to point 1 by reviewer 1.

Figure 1D indicates that 22C10 only neutralized the live virus infection by 50%, unlike the other three RBD-targeting antibodies. Is this a plateau, or can the concentration of 22C10 be increased further to achieve greater neutralization? Interestingly, the authors noted that 22C10 binding to the S1A (NTD) induces conformational changes in the S by causing the S1B to move towards a partially open conformation. This raises the question of whether 22C10 could have a synergistic effect with the other three RBD-targeting antibodies.

Our data indicate that 22C10 neutralized the virus by approximately 50% at the four highest concentrations tested, suggesting that it has indeed reached a plateau in its neutralization capacity. To address the question of whether 22C10 could have a synergistic effect with other RBD-targeting antibodies, we tested whether the partial opening induced by 22C10 exposure is sufficient to expose cryptic epitopes. However, based on our new BLI data (which is consistent with the cryo-EM structure of the 22C10/Spike complex), we found that this conformational shift induced by 22C10 binding is not adequate to facilitate S interaction with mAb 42H3. We included these results in the revised manuscript (line 248-249, Fig.S3C). Based on the inability of 22C10 to expose the cryptic epitopes we do not expect synergy with the other RBD-targeting antibodies to

occur. While exploring such potential synergistic effects of 22C10 presents an intriguing avenue for future research, it necessitates careful experimental design and rigorous testing to accurately assess and quantify any such effects.

Could the authors please explain why the ELISA method could detect the binding of three RBD-targeting antibodies to the prefusion S, while the BLI method could not?

We would like to refer the reviewer to the response provided to point 4 by reviewer 2.

It is also noteworthy that in Figure 3A, the heated S lost its binding with 22C10, but gained binding with three RBD-targeting antibodies. Could the authors explain the loss of 22C10 binding?

We reasoned that the region of PDCoV S that encompasses the 22C10 epitope is more susceptible to disruption by the heat treatment, whereas the RBD has higher thermal stability. This explanation has been added to the results section (Lines 253-254).

In line 238, the number of S1B neutralizers is three, not four.

This has been corrected.

In line 110, Binding kinetics of S-specific mAbs to PDCoV S1B. S1B or S1?

It should be S1, this has been adapted.

REVIEWERS' COMMENTS

Reviewer #1 (Remarks to the Author):

The authors have adequately addressed my comments and the revised manuscript is suitable for publication.

Reviewer #2 (Remarks to the Author):

My concerns have been addressed.

Reviewer #3 (Remarks to the Author):

I am satisfied with the authors' answers to my questions and suggestions. I recommend acceptance for publication.

RESPONSE TO REVIEWERS' COMMENTS

We thank all reviewers for their positive feedback and their assistance during the reviewing process.

Reviewer #1 (Remarks to the Author):

The authors have adequately addressed my comments and the revised manuscript is suitable for publication.

Reviewer #2 (Remarks to the Author):

My concerns have been addressed.

Reviewer #3 (Remarks to the Author):

I am satisfied with the authors' answers to my questions and suggestions. I recommend acceptance for publication.